# Vascular Access for Hemodialysis and Right Ventricular Remodeling: A Prospective Echocardiographic Study

**DOI:** 10.3390/jcm14155565

**Published:** 2025-08-07

**Authors:** Denis Fornazarič, Jakob Gubenšek, Manja Antonič, Marta Cvijić, Jernej Pajek

**Affiliations:** 1Department of Nephrology, University Medical Center Ljubljana, Zaloška 2, 1000 Ljubljana, Slovenia; denis.fornazaric@kclj.si (D.F.); jakob.gubensek@kclj.si (J.G.); 2Faculty of Medicine, University of Ljubljana, Vrazov Trg 2, 1000 Ljubljana, Slovenia; manja.antonic1@gmail.com (M.A.); marta.cvijic@kclj.si (M.C.); 3Department of Dialysis, General Hospital Trbovlje, Rudarska Cesta 9, 1420 Trbovlje, Slovenia; 4Department of Cardiology, Ljubljana University Medical Centre, Zaloška 2, 1000 Ljubljana, Slovenia

**Keywords:** arteriovenous fistula flow, cardiac remodeling, three-dimensional echocardiography, end-stage kidney disease

## Abstract

**Background**: Arteriovenous fistulas (AVFs) may contribute to cardiac remodeling and consequently to an increased risk of heart failure and cardiovascular mortality in patients with end-stage kidney disease (ESKD). We aimed to assess cardiac changes following AVF creation and identify potential parameters associated with cardiac remodeling. **Methods**: In our prospective, single-center study, ESKD patients without significant pre-existing cardiac disease underwent 2D and 3D echocardiographic evaluation before and after AVF creation, along with AVF flow measurement. Cardiac remodeling was assessed using 3D indexed left and right ventricular end-diastolic volumes (LVEDVi, RVEDVi), while systolic function was assessed using longitudinal strain and 3D ejection fraction. **Results**: We included 20 patients (18 men; median age 73.5 years [IQR: 67–77]) with a mean AVF flow of 1140 ± 345 mL/min. At a median of 8.2 months (IQR: 7.3–9.3) following AVF creation, significant biventricular dilatation was observed: LVEDVi increased from 89 ± 14 to 97 ± 21 mL/m^2^ (*p* < 0.05) and RVEDVi from 80 ± 15 to 91 ± 18 mL/m^2^ (*p* < 0.05), while the systolic function of both ventricles did not change significantly. The right ventricle showed the most pronounced remodeling and it was independently associated with volume overload (*p* = 0.003) and elevated left ventricular filling pressure (*p* = 0.030), but not with AVF flow. **Conclusions**: Moderate AVF flow was associated with cardiac remodeling, primarily affecting the right ventricle. Fluid overload and left ventricular filling pressure were key factors associated with right ventricular remodeling, underscoring the need for careful fluid management and vascular access planning in ESKD patients.

## 1. Introduction

The global prevalence of end-stage kidney disease (ESKD) is rising, and hemodialysis remains the most common modality of renal replacement therapy [1]. Despite improvements in overall survival among patients on hemodialysis, their mortality rate remains significantly higher compared to the general population [1,2]. Cardiovascular diseases are the leading cause of death in this patient population, including stroke, myocardial infarction, heart failure (HF), and sudden cardiac death (SCD) [3]. SCD accounts for approximately 34% of all deaths in patients on hemodialysis [4]. It is not only the leading cause of mortality in this population, but may also be directly influenced by ysthe hemodialysis procedure itself, as suggested by recent evidence [5]. The cardiovascular burden in ESKD is significantly increased, with HF being a particularly prominent and multifactorial contributor that remains poorly understood. One of the contributing factors is the presence of vascular access, either an arteriovenous fistula (AVF) or a graft (AVG). Nevertheless, AVF is considered the gold standard for hemodialysis, because it reduces the risk of infections and hospitalizations and need for interventions to maintain patency compared to central venous catheters [6].

The creation of an arteriovenous anastomosis leads to a reduction in systemic vascular resistance and a simultaneous increase in venous return. Neurohormonal activation induces expansion of blood volume, as well as increased heart rate and contractility; these changes can result in elevated cardiac output. Blood flow in the vascular access (Qacc) above 2 L/min or a Qacc-to-cardiac output ratio above 30% defines a high-flow fistula, which is associated with a significantly increased risk of maladaptive cardiac remodeling and heart failure [7,8,9]. However, the vascular access flow threshold that is harmful to cardiac function has not been clearly established, particularly in patients with access flow below 2 L/min or with pre-existing cardiac disease.

The number of longitudinal studies evaluating the effect of AVF on cardiac morphology and function using echocardiography is limited, and there are even fewer that specifically examine the right ventricular (RV) changes. In the short- to mid-term period following AVF creation, studies generally report an increase in cardiac output, left atrial size, and left ventricular (LV) mass and dimensions, while LV systolic function typically remains unchanged or improves [10,11,12]. However, an increase in right heart failure incidence has been observed in a retrospective study [13]. In the long term, mild reduction in LV systolic function may occur, and a higher rate of right heart dilatation and dysfunction is reported [14]. However, factors that determine which patients will develop right heart remodeling after AVF placement remain poorly understood. Moreover, there is a lack of studies employing advanced echocardiographic techniques capable of detecting subclinical changes, such as three-dimensional (3D) volume assessments and strain measurements of the ventricles, which could provide more precise evaluation of cardiac remodeling.

The aim of this prospective study was to evaluate morphological and functional cardiac changes in relation to vascular access flow, using both conventional and advanced echocardiographic parameters during the mid-term period following vascular access creation. We hypothesized that the earliest and most significant remodeling would occur in the right heart chambers and therefore focused our analysis specifically on right ventricular structural and functional alterations. In addition, we sought to identify predictive factors associated with cardiac remodeling. To our knowledge, this is the first prospective study employing 3D echocardiography to assess RV remodeling after AVF creation.

## 2. Materials and Methods

### 2.1. Study Population and Protocol

Our prospective, single-center study consecutively included adult patients with advanced chronic kidney disease or ESKD without significant pre-existing cardiac disease who were referred for the first-time placement of an AVF/AVG. Patients were eligible for inclusion if they did not have any of the following conditions: severe heart failure (LV ejection fraction < 35%), severe valvular or pericardial disease, severe pulmonary hypertension (systolic pulmonary arterial pressure > 60 mmHg), decompensated liver cirrhosis, hyperthyroidism, severe anemia, sepsis, previous heart transplantation, or an episode of acute coronary syndrome or decompensated heart failure within the past month. Patients with poor echocardiographic image quality and vascular access thrombosis or access non-maturation (Qacc < 250 mL/min) during the study were excluded from the analysis.

Baseline echocardiography was performed within 3 months prior to AVF placement, and follow-up echocardiography along with vascular access flow measurements was performed 6 to 9 months after the procedure. Blood pressure and heart rate measurements, bioimpedance spectroscopy (BIS), and medical record review were performed concurrently with both echocardiographic examinations. In patients undergoing chronic hemodialysis all measurements were obtained about 24 h after the most recent dialysis session. In patients who have not yet started chronic hemodialysis, measurements were performed under clinically stable conditions, with no recent changes in diuretic therapy. All assessments, including echocardiography, were conducted on an outpatient basis.

The study protocol was approved by the National medical ethics committee of the Republic of Slovenia (0120-76/2020/3). The study conformed to the principles outlined in the Declaration of Helsinki. Written informed consent was obtained prior to recruitment.

### 2.2. Echocardiography Examination and Data Analysis

Transthoracic echocardiography was performed using the Vivid™ E95 ultrasound system (GE Healthcare, Horten, Norway) with a 2.5–5 MHz transducer. Image acquisition was carried out by one of two experienced examiners (DF, MC). Image analysis was performed offline using EchoPAC software (GE Medical Systems, version 20.6) by both examiners. Measurements were averaged between both examiners. All echocardiographic measurements were performed offline after image acquisition. The examiners were aware of the timing of the echocardiographic acquisitions; however, they were unaware of the baseline echocardiographic assessment when follow-up measurements were performed.

Left and right ventricular assessment was performed using conventional 2D and 3D echocardiography. Cardiac chamber quantification was performed according to the joint recommendations of the American Society of Echocardiography and European Association of Cardiovascular Imaging [15]. LV end-diastolic (LVEDV) and end-systolic volumes (LVESV) and ejection fraction (LVEF) were calculated using the biplane Simpson’s method from apical four- and two-chamber views. LV mass was derived from the 2D cube formula. Left atrial volume (LAV) was measured using the biplane disk summation method. Stroke volume (SV) was calculated as the product of the left ventricular outflow tract (LVOT) area and velocity–time integral (VTI) obtained by pulsed-wave Doppler. Transmitral E and A velocities, E/A ratio, and mitral annular e′ velocities (septal and lateral) were assessed using pulsed and tissue Doppler imaging; the E/e′ ratio was used to estimate LV filling pressures. Maximal tricuspid regurgitation velocity (TRV) was measured by continuous-wave Doppler. Basal RV diameter, tricuspid annular plane systolic excursion (TAPSE), RV fractional area change (FAC), and tricuspid lateral annular systolic velocity (s′) were obtained in the RV-focused modified apical four-chamber view. Right atrial volume (RAV) was measured from the apical four-chamber view using the monoplane area–length method. Systolic pulmonary artery pressure (SPAP) was estimated as 4 × (TRV)^2^ + estimated right atrial pressure (from IVC diameter). SVR was calculated as the difference between mean arterial pressure (MAP) and right atrial pressure, divided by cardiac output, and multiplied by 80 to convert the units to dyn·s·cm^−5^.

Real-time 3D echocardiography was performed using ECG-gated multi-beat acquisition to achieve frame rates of 20–30 frames per second. Three-dimensional LV and RV volumes and ejection fraction were analyzed semi-automatically using EchoPAC software.

LV global longitudinal strain (LVGLS) was assessed by 2D speckle-tracking (Automated Function Imaging (AFI) module) in three apical views and averaged. RV global longitudinal strain (RVGLS) was derived from six segments in the RV-focused modified apical four-chamber view. RV free wall longitudinal strain (RVFWLS) was calculated using the three segments of the RV free wall. Strain measurements were performed according to a previously described methodology [16,17].

### 2.3. Bioimpedance

Extracellular volume excess was assessed using the Body Composition Monitor (BCM^®^, Fresenius Medical Care, Sankt Wendel, Germany), based on whole-body bioimpedance spectroscopy (BIS). Measurements of absolute overhydration (OH) were obtained and used as an estimate of fluid overload.

### 2.4. Vascular Access Flow Measurements

Vascular access blood flow (Qacc) was measured with ultrasound. The diameter (2r) of the brachial artery was measured in a longitudinal plane from inner edge to inner edge. At the same location, a pulsed-wave Doppler signal was acquired with the insonation angle maintained below 60°, and the sample volume was adjusted to encompass nearly the entire vessel lumen to calculate the time-averaged mean velocity. Small adjustments of sample volume position were made to acquire the signal with the lowest amount of turbulent dispersion and from the straight arterial segment. Qacc was calculated using the following formula—time-averaged mean velocity × cross-sectional area (π × r^2^) × 60—with the result expressed in mL/min.

### 2.5. Statistical Analysis

Statistical analysis was performed using IBM SPSS Statistics for Windows, Version 25.0 (IBM Corp., Armonk, NY, USA). Variable distribution was assessed using the Shapiro–Wilk test. Normally distributed data are presented as mean ± standard deviation, and non-normally distributed data as median with interquartile range (IQR). Categorical variables are expressed as frequencies and percentages. For paired comparisons before and after AVF placement, a paired t-test or Wilcoxon signed-rank test was used, depending on data distribution. Univariable linear regression analysis was applied to identify potential predictors of cardiac remodeling following AVF formation. Clinically relevant variables and those with statistically significant *p*-values in univariable analysis were included in a multivariable regression model to determine independent predictors. Collinearity of variables was tested using variance inflation factors (VIFs). To avoid multicollinearity between the univariate variables, multiple models were constructed. The model with the highest log-likelihood values was presented as the final multivariable model. Interobserver variability was assessed using the intraclass correlation coefficient (ICC). A two-tailed *p*-value < 0.05 was considered statistically significant for all analyses.

## 3. Results

### 3.1. Study Population

A total of 29 patients were recruited, of whom 20 completed the study and were included in the final analysis. A total of nine patients were excluded from the analysis: three due to vascular access complications (thrombosis, non-maturation, or ischemia); three due to withdrawal of consent or cancellation of AVF placement; one due to newly diagnosed severe aortic stenosis; one due to an acute cardiovascular event; and one patient who died before follow-up echocardiographic examination. Baseline characteristics of the study cohort are presented in Table 1. At inclusion, five patients (25%) were on maintenance hemodialysis (as their first dialysis modality) using jugular hemodialysis catheters; none were receiving peritoneal dialysis, while the others had advanced chronic kidney disease.

Regarding the baseline echocardiographic characteristics of the study cohort, the median LVEF was 54% (IQR: 53 to 56). Two patients had EF below 50% (38% and 45%). Elevated left ventricular filling pressure was observed in eight (40%) patients; pulmonary hypertension (SPAP ≥ 35 mmHg, estimated by echocardiography) was present in six (30%) patients.

The median time from baseline echocardiography to vascular access creation was 6 (IQR: 1 to 15) days, while follow-up echocardiography and fistula flow measurements were performed 8.2 (IQR: 7.3 to 9.3) months after surgery. The mean vascular access flow of the study cohort was 1140 ± 345 mL/min (range: 711–1868 mL/min) and the mean Qacc/CO was 19.7 ± 6%. Distal access (radio-cephalic AVF) was created in 11 patients, while 9 patients had proximal vascular access, including 4 with AVG. Vascular access flow distributions are shown in Figure 1. None of the patients met the criteria for a high-flow arteriovenous fistula.

### 3.2. Hemodynamic Changes After AVF Placement

Evaluation of hemodynamic parameters before and after vascular access placement is reported in Table 2. Arterial blood pressure as well as SPAP did not change significantly. Stroke volume significantly increased after AVF placement, while no significant changes were observed in cardiac output; none of the patients developed high cardiac output (>3.9 L/min/m^2^). Echocardiographic assessment of CVP was comparable before and after AVF placement. Also, measurement of absolute overhydration using BCM was similar before and after AVF placement (2.0 ± 2.1 L vs. 1.3 ± 2.1 L, *p* = 0.231).

### 3.3. Morphological and Functional Cardiac Changes After AVF Placement

Pairwise comparisons of echocardiographic parameters demonstrated a significant increase in the volume of all cardiac chambers (Figure 2). The right ventricle showed the most pronounced remodeling. Indexed 3D RVEDVi increased by 19 ± 27%, while 3D LVEDVi increased by 11 ± 19%, LAVI by 13 ± 21%, and RAVI by 14 ± 24%. A significant increase in RV volume, defined as a ≥20% rise in 3D RVEDVi, was observed in 35% of patients (7 out of 20). Conventional 2D echocardiographic parameters of LV dimension and geometry showed no significant changes after AVF creation (Table 3). However, 2D RV basal diameter increased during follow-up.

Echocardiographic parameters considered to reflect ventricular function, both systolic and diastolic function, showed nonsignificant changes after AVF placement (Table 3). None of the patients developed RV systolic dysfunction, defined by a 3D RVEF of less than 40%, after vascular access placement.

### 3.4. Parameters Associated with Cardiac Remodeling After AVF Placement

Right ventricular dilatation was the main clinical and statistically significant change observed in our cohort. To identify parameters associated with the increase in RVEDVi after AVF placement, we performed a univariate linear regression analysis (Table 4). Baseline 3D RVEDVi, 3D RVEF, RVGLS, RVFWLS, E/e′, presence of coronary artery disease (CAD), and follow-up overhydration were significantly associated with the increase in RV volume. In the multivariate regression model, only follow-up overhydration (β = 0.533, *p* = 0.003), E/e′ (β = 0.283, *p* = 0.030) and the presence of CAD (β = 0.313, *p* = 0.038) remained significantly associated with RV dilatation after AVF placement (Appendix A). In contrast, vascular access flow did not show significant association with the degree of RV dilatation.

When follow-up overhydration was added to a multivariate model with baseline characteristics including E/e′, CAD, RVFWLS, and RVEDVi, the adjusted R^2^ increased from 0.523 to 0.675 (*p* = 0.021), confirming its independent and additive predictive value (Figure 3).

### 3.5. Reproducibility

Interobserver variability for cardiac chamber measurements (3D LVEDV, 3D LVESV, 3D RVEDV, 3D RVESV) and systolic function parameters (3D LVEF, 3D RVEF, LVGLS, RVGLS, RVFWL) showed good to excellent agreement. Intraclass correlation coefficients (ICC) ranged from the lowest of 0.77 (95% CI 0.40–0.91) for 3D RVEF to the highest of 0.99 (95% CI 0.88–0.997) for 3D LVESV.

## 4. Discussion

In this prospective study, we assessed longitudinal morphological and functional cardiac changes during the mid-term period following AVF placement. To the best of our knowledge, this is the first prospective study that comprehensively assessed both the left and right ventricles using advanced 3D echocardiographic parameters and strain measurements in patients undergoing vascular access placement. We found a significant increase in the volumes of all cardiac chambers, with the most pronounced remodeling observed in the right ventricle. Importantly, these structural changes occurred without accompanying impairment of systolic function in either ventricle. Furthermore, we identified preoperative left ventricular filling pressure (E/e′) and fluid overload at follow-up as potential independent predictors of RV dilatation. In contrast, within the range of AV access flows observed in our cohort, access flow itself was not independently associated with remodeling.

### 4.1. Hemodynamic Changes After AVF Placement

Patients with ESKD are prone to volume overload, which can itself significantly affect cardiac hemodynamics. The placement of an AVF can lead to further hemodynamic changes, which in turn act as a systemic shunt. This can result in a decrease in systemic vascular resistance and unloading of the left ventricle, while an increase in venous return can overload the right ventricle [12]. However, in our study, we did not observe significant reduction in systemic vascular resistance or arterial blood pressure, nor an increase in cardiac output or SPAP. This might be due to the relatively low AVF flow and adequate volume control after AVF formation in our study cohort compared to studies where such hemodynamic changes were noted [10,12].

### 4.2. Morphological and Functional Cardiac Changes After AVF Placement

Previous studies have shown inconsistent results regarding changes in LV size and function after AVF placement [10,11,12,14]. While studies have uniformly reported RV dilatation, the degree of RV dysfunction was variable [13,14]. These studies included patients with a wide range of AVF flow rates. The older studies mainly included patients with high-flow vascular access [8,18,19], which may explain some of the discrepancies in the results. Additionally, most of these studies are retrospective, using mainly 2D echocardiography and linear measurements of heart chambers, such as end-diastolic diameter, and conventional parameters for assessing systolic function, such as EF. However, 3D echocardiography enabled a more accurate and reproducible assessment of ventricular volumes and EF compared to 2D echocardiography [15].

From a pathophysiological perspective, changes in loading conditions are often accompanied by changes in chamber shape and function through the process of remodeling [20,21,22]. Initially, during the adaptive remodeling phase, an increase in venous return is normally coupled with an increase in chamber dimensions that augment stroke volume through the Frank–Starling mechanism. Increased preload augments myocardial contraction, as long as contractility is normal and there is no significant increase in afterload. However, a long-lasting increase in excessive venous return results in a vicious cycle of progressive chamber dilatation, which leads to an increase in wall stress and consequently an increase in afterload due to increased myocardial tension required for ejection. This eventually results in contractile dysfunction, which represents the transition to maladaptive remodeling. Although there is no consensus on what is a safe vascular access flow from the cardiac perspective, high-flow access is often linked with maladaptive remodeling and signs and symptoms of heart failure, though not exclusively [23,24]. Maladaptive remodeling in ESKD patients would typically present with deterioration of systolic function, while the subclinical phase could be detected through abnormal ventricular strain parameters [25,26]. Impaired LVGLS in dialysis patients was independently associated with higher rates of heart failure hospitalization and all-cause mortality, despite preserved LVEF [26,27]. Similarly, impaired RV strain was an independent predictor of right heart failure following AV access creation [13]. In our study cohort, cardiac remodeling affected both the atria and the ventricles, with the most pronounced dilatation occurring in the right ventricle. The anatomical differences between the two ventricles make the right ventricle more susceptible to loading changes and a greater extent of remodeling [20]. Comparing the extent of ventricular dilatation after vascular access placement in our study with the existing literature is challenging, as few studies have comprehensively assessed chamber volumes, and ours is the first to do so using 3D echocardiography. We observed a greater increase in RV volume compared to LV volume (19% vs. 11%). Despite these structural changes, both systolic and diastolic ventricular function remained preserved. These findings suggest that AVF creation, when flow rates are within the target range as observed in our cohort, induces physiological adaptation or adaptive remodeling.

Most studies that have evaluated RV changes after vascular access placement relied on semiquantitative assessments of volume and systolic function and retrospective study design [13,14,19,28,29,30]. These studies commonly reported RV enlargement accompanied by systolic dysfunction. Gumus et al. observed overt RV failure in 18.5% of patients at a mean follow-up of 2.1 months postoperatively [13], while Reddy et al. reported worsening RV function in 34% of patients at a median follow-up of 2.5 years [14]. However, in a study by Gumus et al. RV systolic function was impaired in a substantial proportion of the study cohort already before vascular access placement, while in the study by Reddy et al. data on AV flow were not available, but the authors reported high cardiac output indices (>4 L/min/m^2^) in 21% of patients and pre-existing RV dysfunction in 12%. In contrast, RV systolic function in our cohort remained normal despite a significant increase in RV volume. This may be explained by several factors. Firstly, baseline RV function was preserved, with normal ejection fraction as well as strain values. Secondly, the AVF flow was within a moderate range (mean 1140 ± 345 mL/min) and not excessive, and therefore did not result in a substantial increase in cardiac output or SPAP. Our results indicate that for patients with a healthy right ventricle, a moderate AVF flow is actually safe as it did not significantly affect loading conditions or induce maladaptive RV remodeling during a median follow-up period of 8 months. However, our results cannot be extrapolated to longer follow-up periods, as AVF flow may continue to increase months or even years after placement due to vascular remodeling, particularly in proximal brachial artery-based access [29]. In contrast, remodeling of the radial artery is relatively short, typically lasting 3 to 6 weeks [31]. Furthermore, radial artery-based AVF blood flow is generally lower from the onset compared to that of proximal AVFs, which are more prone to development of high flow and heart failure [7]—although this was not observed in our cohort, probably due to the low number of patients and relatively short duration of observation. Nevertheless, even in the absence of additional risk factors, persistently elevated preload, initially manifesting as physiological remodeling, as seen in our study, may progress to maladaptive changes due to the vicious cycle of progressive chamber enlargement with increasing wall stress and eventually deterioration of systolic function [20]. Therefore, further long-term studies are needed to assess this problem. As it is unclear who is at increased risk of developing maladaptive remodeling after AVF creation, we believe that regular echocardiographic follow-up might be necessary even in cases where only physiological remodeling is initially present.

### 4.3. Parameters Associated with Cardiac Remodeling After AVF Placement

Although our study cohort demonstrated presumably adaptive or physiological cardiac remodeling, long-lasting volume overload can result in progressive ventricular enlargement and eventually contractile dysfunction [20]. Therefore, it is important for clinical practice to identify a subset of patients who are at increased risk of developing remodeling. Our findings suggest that fluid overload and LV filling pressure were associated with RV dilatation. Preserved RV geometry is essential for maintaining normal LV function and vice versa. RV dilatation can impair LV filling through diastolic leftward septal shift. On the other hand, elevated LV filling pressure may mitigate this septal shift, thereby helping to preserve both LV and RV function. It has been estimated that more than half of RV systolic work is supported by LV contraction through ventricular interdependence [32]. These interactions highlight the importance of balanced ventricular loading conditions in patients undergoing AVF placement, including avoiding excessive preload increases from access flow and ensuring adequate volume control in order to minimize the risk of maladaptive cardiac remodeling.

In this context, dialysis modality and ultrafiltration rate may play a significant role in modulating cardiac remodeling. Prior randomized studies have demonstrated that daily dialysis leads to improved volume and blood pressure control, reduced interdialytic weight gain, and lower ultrafiltration volumes or rates compared to conventional thrice-weekly schedules [33,34]. These changes were associated with a significantly greater reduction in left ventricular mass (−16.4 ± 2.9 g vs. −2.6 ± 3.2 g), and reductions in both LVEDV and RVEDV (by 11% and 12%, respectively), with more pronounced effects observed in anuric patients. Even after a single hemodialysis session, preload reduction can significantly decrease LVEDV, and the right ventricle appears to be even more sensitive to such preload changes [35]. Tailored strategies—such as frequent or prolonged dialysis sessions aimed at avoiding excessive ultrafiltration rates and achieving stricter volume control—may therefore be especially beneficial in mitigating cardiac remodeling following AVF creation.

### 4.4. Limitations

Our study has several limitations. First, the sample size was relatively small, which limits the statistical power and generalizability of our findings. Second, our cohort was predominantly male, with 90% of participants being men. Sex-related differences in cardiac structure and function are well established; women typically have smaller ventricular volumes and higher baseline systolic function, suggesting a more hyperdynamic cardiovascular profile. This could potentially result in a different remodeling response after AVF placement; however, further research is needed to evaluate gender diversity. Additionally, our study cohort comprised mainly patients with preserved ejection fraction. It is not known whether the extent of the remodeling after AVF placement would be comparable in patients with reduced ejection fraction or any other underlying cardiac diseases. When systolic dysfunction is already present, some compensatory mechanisms have already been activated, making the heart more susceptible to small changes in preload as well as afterload. This could lead to maladaptive hemodynamic compensation and progression to heart failure earlier or at lower AVF flow rates. Third, the study lacked a control group of ESKD patients without AVF—such as those on peritoneal dialysis or using a central venous catheter—making it more difficult to isolate the impact of vascular access creation on cardiac remodeling. Fourth, the variability of vascular access flow rates was limited, with no patients developing high-flow AVF, which may have reduced our ability to detect flow-dependent effects. Finally, the follow-up period of approximately 8 months may be insufficient to detect the late development of maladaptive cardiac remodeling, due to persistently elevated cardiac preload and potential increases in AVF flow over time. However, our study population reflects current standards for vascular access creation, aiming to achieve sufficient blood flow for dialysis, while minimizing the risk of high-flow complications. Additionally, most patients had preserved RV function and relatively well-maintained LV function, aside from diastolic dysfunction, potentially restricting the applicability of our findings to more diverse patient populations.

Taken together, these limitations underscore the need for larger, long-term prospective studies with more balanced gender representation, a wider range of underlying cardiac dysfunction, a broader spectrum of vascular access flow rates, and appropriate control groups. It would also be valuable to determine whether early cardiac remodeling after AVF placement can predict long-term cardiac changes or even clinical outcomes. Future research should also ensure careful assessment of potential confounding factors—such as fluctuations in fluid overload and dialysis modality—to better characterize the impact of vascular access placement in ESKD patients and to identify individuals at higher risk for maladaptive cardiac remodeling.

## 5. Conclusions

In this prospective study involving ESKD patients and employing advanced echocardiographic parameters, we observed significant biventricular volumetric remodeling following AVF placement, without evidence of overt or subclinical systolic dysfunction. Right ventricular dilatation, objectively quantified by 3D echocardiography, was independently associated with elevated left ventricular filling pressure and volume overload. In contrast, access flow—maintained within the moderate range—was not a significant predictor of remodeling. These findings underscore the importance of rigorous volume management in this population and suggest that careful planning of vascular access to prevent the development of high-flow states is essential.

## Figures and Tables

**Figure 1 jcm-14-05565-f001:**
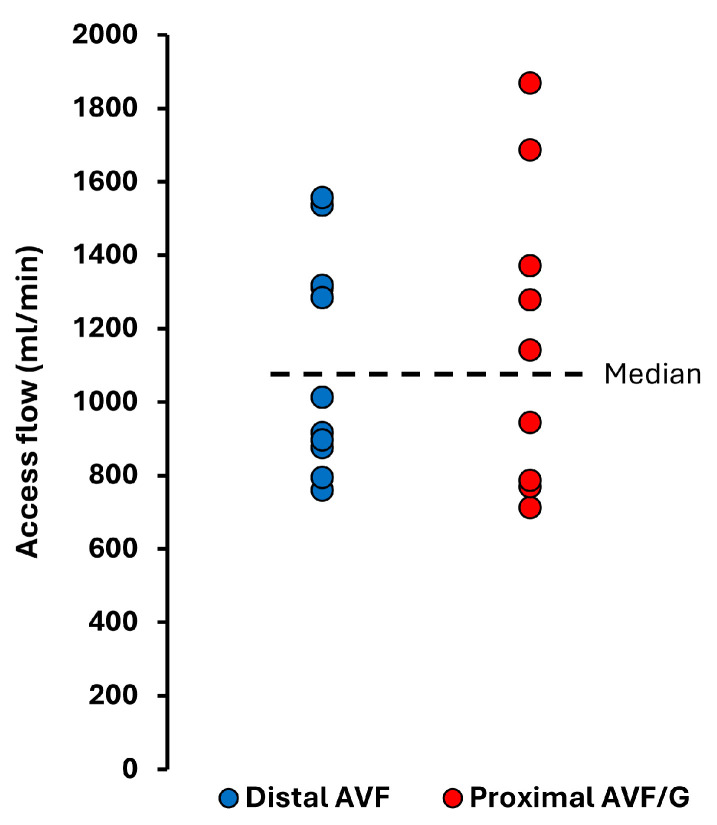
Vascular access flow in the study cohort at a median of 8.2 months after vascular access creation.

**Figure 2 jcm-14-05565-f002:**
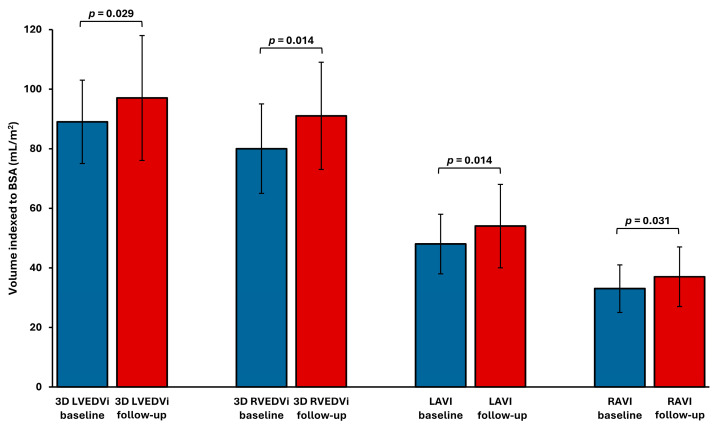
Cardiac chamber volume indices before and after AVF placement. LAVI—left atrial volume index; LVEDVi—left ventricular end-diastolic volume index; RAVI—right atrial volume index; RVEDVi—right ventricular end-diastolic volume index.

**Figure 3 jcm-14-05565-f003:**
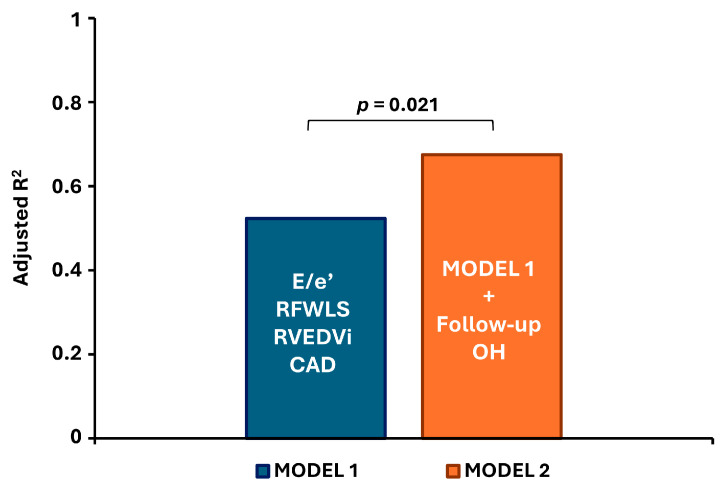
Contribution of follow-up overhydration to the predictive model for right ventricular dilatation. All parameters in Model 1 represent baseline values. CAD—coronary artery disease; E/e′—ratio of early mitral inflow velocity to early diastolic mitral annular velocity; RFWLS—right ventricular free wall longitudinal strain; RVEDVi—right ventricular end-diastolic volume index; OH—overhydration.

**Table 1 jcm-14-05565-t001:** Baseline characteristics of the study cohort.

**Demographics**	
Age, years	73.5 (IQR: 67 to 77)
Male, *n* (%)	18 (90)
Body mass index	26.1 ± 3.5
**Comorbidities**	
Coronary artery disease, *n* (%)	4 (20)
Diabetes mellitus, *n* (%)	8 (40)
Arterial hypertension, *n* (%)	19 (95)
Hyperlipidemia, *n* (%)	11 (55)
Atrial fibrillation, *n* (%)	1 (5)
**ESKD etiology**	
Hypertensive, *n* (%)	4 (20)
IgA, *n* (%)	4 (20)
Diabetic, *n* (%)	4 (20)
Uknown, *n* (%)	4 (20)
ANCA vasculitis, *n* (%)	2 (10)
Obstructive nephropathy, *n* (%)	2 (10)
ADPKD, *n* (%)	5 (25)
Chronic hemodialysis	5 (25)

ADPKD—autosomal dominant polycystic kidney disease, ANCA—antineutrophil cytoplasmic antibody, ESKD—end-stage kidney disease.

**Table 2 jcm-14-05565-t002:** Hemodynamic parameters before and after AVF placement.

Hemodynamic Parameters	Baseline	Follow-Up	*p*-Value
Systolic BP, mmHg	154 ± 23	152 ± 22	0.780
Diastolic BP, mmHg	85 ± 10	79 ± 15	0.084
Heart rate, bpm	64 ± 9	63 ± 10	0.563
SV Doppler, mL	89 ± 12	94 ± 12	0.004
CO, L/min	5.8 ± 1.0	5.9 ± 0.8	0.621
CI, L/min/m^2^	3.1 ± 0.6	3.1 ± 0.5	0.519
CVP, mmHg	3 (3 to 8)	3 (3 to 8)	0.502
SPAP, mmHg	31 (27 to 42)	40 (33 to 44)	0.214
SVR, dyn·s·cm^−5^	1451 ± 301	1347 ± 147	0.185
Overhydration (BCM), L	2.0 ± 2.1	1.3 ± 2.1	0.231

Values are presented as mean ± SD for normally distributed variables, and median (IQR) for skewed variables. *p*-values are derived from paired *t*-test or Wilcoxon signed-rank test, as appropriate. BCM—Body Composition Monitor; CI—cardiac index; CO—cardiac output; CVP—central venous pressure; SPAP—systolic pulmonary artery pressure; SV—stroke volume; SVR—systemic vascular resistance: (MAP − CVP)/CO × 80.

**Table 3 jcm-14-05565-t003:** Morphological and functional cardiac parameters before and after AVF placement.

LV Dimension and Geometry	Baseline	Follow-Up	*p*-Value
LVEDD, cm	5.2 ± 0.5	5.3 ± 0.6	0.117
LVMI, g/m^2^	96 (81 to 117)	109 (77 to 126)	0.147
2D LVEDV, mL	158 (134 to 168)	159 (142 to 188)	0.083
2D LVEDVi, mL/m^2^	81 (72 to 89)	84 (78 to 99)	0.067
2D LVESV, mL	68 (59 to 78)	69 (60 to 93)	0.095
3D LVEDV, mL	165 (148 to 181)	176 (144 to 206)	0.039
3D LVEDVi, mL/m^2^	89 ± 14	97 ± 21	0.029
3D LVESV, mL	77 (65 to 84)	79 (65 to 103)	0.072
**LV systolic parameters**			
2D LVEF, %	55 (53 to 58)	55 (51 to 57)	0.247
3D LVEF, %	54 (53 to 56)	55 (51 to 56)	0.602
LVGLS, %	−17 ± 2.1	−16.1 ± 1.6	0.132
LV s’, cm/s	7.7 ± 1.4	7.5 ± 1.3	0.493
**LV diastolic parameters**			
E/A	0.9 (0.7 to 1.2)	0.8 (0.6 to 1.3)	0.055
E/e′	11.5 (9 to 15)	10.5 (8.3 to 15.5)	0.407
**RV dimension**			
RV basal diameter, cm	4.6 ± 0.5	4.8 ± 0.4	0.005
RV diastolic area, cm^2^	26.3 ± 3.3	27.4 ± 4.5	0.188
3D RVEDV, mL	134 (128 to 177)	167 (141 to 197)	0.028
3D RVEDVi, mL/m^2^	80 ± 15	91 ± 18	0.014
3D RVESV, mL	61 (53 to 82)	75 (60 to 97)	0.028
**RV systolic parameters**			
3D RVEF, %	56 (54 to 61)	57 (50 to 58)	0.081
RVGLS, %	−21.9 ± 3.2	−21.5 ± 3.1	0.497
RVFWLS, %	−26.6 ± 4.2	−26.1 ± 3.7	0.593
FAC, %	47.7 ± 6.4	46.4 ± 6.4	0.24
TAPSE, mm	27 ± 4	26 ± 5	0.772
RV s’, cm/s	14.9 ± 2.5	14.5 ± 2	0.462

Values are presented as mean ± SD for normally distributed variables, and median (IQR) for skewed variables. *p*-values are derived from paired *t*-test or Wilcoxon signed-rank test, as appropriate. E/A—early to late diastolic transmitral flow velocity ratio; E/e′—ratio of early mitral inflow velocity to early diastolic mitral annular velocity; FAC—fractional area change; LVEDD—left ventricular end-diastolic diameter; LVEDVi—left ventricular end-diastolic volume index; LVEF—left ventricular ejection fraction; LVESV—left ventricular end-systolic volume; LVGLS—left ventricular global longitudinal strain; LVMI—left ventricular mass index; LV s’—left ventricular systolic mitral annular velocity; RVEDVi—right ventricular end-diastolic volume index; RVEF—right ventricular ejection fraction; RVESV—right ventricular end-systolic volume; RVFWLS—right ventricular free wall longitudinal strain; RVGLS—right ventricular global longitudinal strain; RV s’—right ventricular systolic tricuspid annular velocity; TAPSE—tricuspid annular plane systolic excursion.

**Table 4 jcm-14-05565-t004:** Predictors of right ventricular dilatation after AVF placement.

Regression Variable	Univariate Analysis		
	β	95% CI	*p*-Value
Age	0.201	−0.007 to 0.015	0.425
Coronary artery disease	0.604	0.121 to 0.679	0.008
Diabetes mellitus	0.315	−0.102 to 0.442	0.203
Baseline—OH	−0.459	−0.121 to 0.001	0.055
Follow-up—OH	0.656	0.034 to 0.140	0.003
Qacc	0.152	−0.257 to 0.561	0.443
**Baseline RV parameters**			
3D RVEDVi	−0.569	−0.018 to −0.002	0.014
3D RVEF	0.631	0.13 to 0.061	0.005
RVFWLS	−0.584	−0.066 to −0.01	0.011
RVGLS	−0.501	−0.081 to −0.004	0.034
FAC	0.167	−0.015 to 0.029	0.507
TAPSE	0.344	−0.01 to 0.052	0.162
SPAP	−0.371	−0.019 to 0.003	0.130
**Baseline LV parameters**			
LVMI	0.083	−0.004 to 0.006	0.744
3D LVEDVi	−0.260	−0.015 to 0.005	0.313
3D LVEF	0.202	−0.018 to 0.040	0.437
LVGLS	0.047	−0.061 to 0.073	0.854
E/e′	0.497	0.002 to 0.046	0.036

β—standardized regression coefficient; CI—confidence interval; E/e′—ratio of early mitral inflow velocity to early diastolic mitral annular velocity; FAC—fractional area change; LVEDVi—left ventricular end-diastolic volume index; LVEF—left ventricular ejection fraction; LVGLS—left ventricular global longitudinal strain; LVMI—left ventricular mass index; OH—overhydration (from Body Composition Monitor); Qacc—vascular access blood flow; RVEDVi—right ventricular end-diastolic volume index; RVEF—right ventricular ejection fraction; RVFWLS—right ventricular free wall longitudinal strain; RVGLS—right ventricular global longitudinal strain; SPAP—systolic pulmonary artery pressure; TAPSE—tricuspid annular plane systolic excursion.

## Data Availability

The raw data supporting the conclusions of this article will be made available by the authors on request.

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
