# Peer review of "Vascular Access for Hemodialysis and Right Ventricular Remodeling: A Prospective Echocardiographic Study"

_jcm, 2025, doi:10.3390/jcm14155565_

Round 1

Reviewer 1 Report

Comments and Suggestions for Authors

Dear Authors,

I commend you on this well-conducted and clearly presented prospective study exploring the impact of arteriovenous fistula (AVF) creation on cardiac remodeling, particularly of the right ventricle, in patients with end-stage kidney disease (ESKD). The use of advanced echocardiographic parameters, including 3D imaging and strain analysis, is a clear strength and adds robustness to your findings.

That said, I offer the following suggestions to improve clarity and impact:

Introduction:

The background is well framed, but consider clarifying the novelty of your study more explicitly. You mention that this is the first prospective study using 3D echocardiography to assess RV remodeling post-AVF. Emphasizing this point earlier could strengthen the rationale.

Methods:

-Please clarify the time window between the last dialysis session and follow-up echocardiography in all patients, especially those not yet on chronic hemodialysis.

-In the echocardiographic section, indicate if examiners were blinded to the timepoint (pre/post AVF), to minimize bias in volume measurements.

Results:

-Table 4 could benefit from clearer formatting: bolding statistically significant multivariable predictors would help the reader quickly identify key results.

-Consider reporting the proportion of patients who had significant changes in volume status or RVEDVi (e.g., Δ > 10%), not just mean values.

Discussion:

-The distinction between adaptive and maladaptive remodeling is well addressed. However, given the modest sample size and short follow-up, it may be helpful to more strongly acknowledge the risk of long-term maladaptation, particularly in patients with increasing AVF flow over time.

-Please consider discussing the potential role of AVF location (distal vs. proximal) in remodeling patterns, as this was briefly mentioned but not deeply explored.

Language and style:

Overall, the manuscript is well written. Minor improvements in phrasing would enhance flow and precision. (e.g., "volume status" could be replaced with “fluid overload” or "extracellular volume excess" in some contexts to be more physiologically accurate).

Author Response

Response to Reviewers

We sincerely thank you for your time and valuable suggestions for improving our manuscript. We also appreciate the prompt and concise feedback, as well as the clear and constructive comments. We have made every effort to revise the manuscript as accurately and thoroughly as possible.

REVIEWER 1:

Point-by-point response to Comments and Suggestions for Authors

Comment 1: The background is well framed, but consider clarifying the novelty of your study more explicitly. You mention that this is the first prospective study using 3D echocardiography to assess RV remodeling post-AVF. Emphasizing this point earlier could strengthen the rationale.

Response 1: To better clarify the aim of our study and emphasize its originality, we have revised the last paragraph of the Introduction to include the following sentence (p. 3, line 97-98): To our knowledge, this is the first prospective study employing 3D echocardiography to assess RV remodeling after AVF creation.

Comment 2.1: Please clarify the time window between the last dialysis session and follow-up echocardiography in all patients, especially those not yet on chronic hemodialysis.

Response 2.1: Thank you for pointing out this inconsistency. We have added a clarification regarding the timing of follow-up measurements (including echocardiography) in patients not yet on dialysis. The revised sentence (p. 3, line 117-120): In patients, who have not yet started chronic hemodialysis, measurements were performed under clinically stable condition, with no recent changes in diuretic therapy. All assessments, including echocardiography, were conducted on an outpatient basis.

Comment 2.2: In the echocardiographic section, indicate if examiners were blinded to the timepoint (pre/post AVF), to minimize bias in volume measurements.

Response 2.2: Thank you for raising this important comment. We tried to minimize the bias in measurements as much as possible. All echocardiographic measurements were performed offline after image acquisition. The examiners were familiar with the timing of the echocardiographic assessment (data available from the time information in echo clips). However, when performing follow-up echo measurements, examiners were unaware with the baseline measurements. Importantly, 3D volumes assessments and strain measurements were performed using semi-automatic echocardiographic software (already stated in the manuscript), which have been proven to improve reproducibility and reduce bias when compared with purely manual methods. Therefore, we believe that our results are based on objective and reliable measurements.  We have added the following explanation in the Methods section (p. 3, line 129-132): All echocardiographic measurements were performed offline after image acquisition. The examiners were aware of the timing of the echocardiographic acquisitions; however, they were unaware of the baseline echocardiographic assessment when follow-up measurements were performed.

Comment 3.1: Table 4 could benefit from clearer formatting: bolding statistically significant multivariable predictors would help the reader quickly identify key results.

Response 3.1: Based on the suggestion of the second reviewer, Table 4 was split, with the multivariate analysis moved to the Supplementary Material as Supplementary Table 1. All significant parameters have been bolded for clarity (p. 9).

Comment 3.2: Consider reporting the proportion of patients who had significant changes in volume status or RVEDVi (e.g., Δ > 10%), not just mean values.

Response 3.2: Thank you for this suggestion. Because the overhydration assessed by body composition monitor did not change significantly in our study population from baseline to follow-up, we have added only the results for changes in RVEDVi to the manuscript. We used a 20% increase in RVEDVi to define significant right ventricular volume increase. Due to the lack of robust data from the literature on the cut-off values for significant changes in RVEDV related to adverse RV remodeling, we used data from the studies focusing on adverse LV remodeling in different cardiac diseases.  Based on the literature, a 10-20 % increase in LVEDV has been proposed as a relevant threshold. Taking into account the physiological and anatomical characteristics of right ventricle, which is more compliant than left ventricle and can handle larger volume changes with relatively smaller increases in pressure, we opted for higher cut-off value. We added the following sentence to results section (p. 7, line 246-248): A significant increase in RV volume, defined as ≥20% rise in RVEDVi, was observed in 35% of patients (7 out of 20).

Comment 4.1: The distinction between adaptive and maladaptive remodeling is well addressed. However, given the modest sample size and short follow-up, it may be helpful to more strongly acknowledge the risk of long-term maladaptation, particularly in patients with increasing AVF flow over time.

Response 4.1: Thank you for noting this important issue. We additionally emphasized that even under the same conditions as in our study, a longer duration of the observation period may lead to different outcomes, even in the absence of additional risk factors—let alone in the presence of increasing flow, as you pointed out. We added the following statement to the discussion section of the manuscript (p. 12, line 422-436):  However, our results cannot be extrapolated to longer follow-up periods, as AVF flow may continue to increase months or even years after placement due to vascular remodeling, particularly in proximal, brachial artery-based access [29]. In contrast, remodeling of the radial artery is relatively short, typically lasting 3 to 6 weeks [31]. Furthermore, radial artery based AVFs blood flow is generally lower from the onset compared to proximal AVFs, which are more prone to development of high flow and heart failure [7]—although this was not observed in our cohort, probably due to low number of patients and relatively short duration of observation. Nevertheless, even in the absence of additional risk factors, persistently elevated preload, initially manifesting as physiological remodeling, as seen in our study, may progress to maladaptive changes due to vicious cycle of progressive chamber enlargement with increasing wall stress and eventually deterioration of systolic function [20]. Therefore, further long-term studies are needed to better assess this problem. As it is unclear who is at increased risk of developing maladaptive remodeling after AVF creation, we believe that regular echocardiographic follow-up might be necessary even in cases where only physiological remodeling is initially present.

Comment 4.2: Please consider discussing the potential role of AVF location (distal vs. proximal) in remodeling patterns, as this was briefly mentioned but not deeply explored.

Response 4.2: We have emphasized the difference in remodeling duration between the radial and brachial arteries in the discussion, as well as the general predisposition for higher flow in proximal AVFs (p. 12, line 425-429): In contrast, remodeling of the radial artery is relatively short, typically lasting 3 to 6 weeks [31]. Furthermore, in radial artery based AVFs blood flow is generally lower from the onset compared to proximal AVFs, which are more prone to development of high flow and heart failure [7] — although this was not observed in our cohort, probably due to low number of patients and short duration of observation.  

Comment 5: Overall, the manuscript is well written. Minor improvements in phrasing would enhance flow and precision. (e.g., "volume status" could be replaced with “fluid overload” or "extracellular volume excess" in some contexts to be more physiologically accurate).

Response 5: The wording has been corrected as suggested.

Reviewer 2 Report

Comments and Suggestions for Authors

This is a very well written manuscript overall. The topic is also of interest. However, due to the very small sample size, this study might be considered as "concept generation" and not a strong "association evidence". My suggestions are included as below: 

1. Introduction

Lines 38-43. You might want to mention about sudden cardiac death which occupies 25% death in patients with hemodialysis. Recent study demonstrating hemodialysis is also a potential contributor for SCD (PMCID: PMC12136628 DOI: 10.34067/KID.0000000705). I strongly believe including those studies might improve the context of your study. 

2. Methods

2.1 Could you clarify that whether any patients receive peritoneal dialysis before hemodialysis? or this is their very first dialysis? 

2.2 If the sample size is too small (<30) in this case, multivariable regression become unreliable. You might consider to leave the multivariable regression findings out of the main text. A place in supplemental materials is more appropriate.

3. Discussion. 

3.1 What do you think about the effect of schedule and duration of hemodialysis on RV changes? Prior studies suggest a daily but short session might produce better outcome compared to standard 3h session x 3 times per weeks? 

3.2 What do you think about the effect of ultrafiltration rate in RV dimension change? 

3.3 What are your suggestion to potentially address the RV/LV dimension changes and blood volume overload in those patients? 

Author Response

Response to Reviewers

We sincerely thank you for your time and valuable suggestions for improving our manuscript. We also appreciate the prompt and concise feedback, as well as the clear and constructive comments. We have made every effort to revise the manuscript as accurately and thoroughly as possible.

REVIEWER 2:

Point-by-point response to Comments and Suggestions for Authors

Comment 1: Lines 38-43. You might want to mention about sudden cardiac death which occupies 25% death in patients with hemodialysis. Recent study demonstrating hemodialysis is also a potential contributor for SCD (PMCID: PMC12136628 DOI: 10.34067/KID.0000000705). I strongly believe including those studies might improve the context of your study. 

Response 1: We fully agree that sudden cardiac death is an essential aspect of the cardiovascular burden in hemodialysis patients and should be acknowledged accordingly. In response to your suggestion, we have now included the following statement in the introduction to reflect this important issue (p. 2, line 44-47): SCD accounts for approximately 34% of all deaths in patients on hemodialysis [4]. It is not only the leading cause of mortality in this population, but may also be directly influenced by the hemodialysis procedure itself, as suggested by recent evidence [5].

Comment 2.1: Could you clarify that whether any patients receive peritoneal dialysis before hemodialysis? or this is their very first dialysis? 

Response 2.1: To provide better clarity, we have added the following sentence (p. 5, line 209-210): At inclusion, five patients (25%) were on maintenance hemodialysis (as their first dialysis modality) using jugular hemodialysis catheters; none were receiving peritoneal dialysis, while the others had advanced chronic kidney disease.

Comment 2.2: If the sample size is too small (<30) in this case, multivariable regression become unreliable. You might consider to leave the multivariable regression findings out of the main text. A place in supplemental materials is more appropriate.

Response 2.2: Thank you for raising this important comment. We fully acknowledge that with n = 20, such models are at risk of overfitting and should be interpreted with caution. However, we believe the multivariable analysis provides valuable exploratory insight that complements the univariable findings. To address this concern, we have taken the following steps: we have clearly described the limitations of the multivariable model in the Discussion section; we restricted the number of variables included based on clinical relevance and prior literature, following "events per variable" (EPV) principles as closely as feasible; the results are explicitly presented as hypothesis-generating, not confirmatory. Given the exploratory nature of this analysis and its added value in identifying potential independent associations, we kindly request to retain the univariate findings in the main text. However, in response to your suggestion, we have moved the multivariable portion of Table 4 to the Supplementary Materials for improved clarity and transparency. The following sentence in the Limitations section highlights the downsides of the multivariable model. (p. 13, line 478-479): First, the sample size was relatively small, which limits the statistical power and generalizability of our findings, especially the multivariable model.

Comment 3.1-3.3:  3.1 What do you think about the effect of schedule and duration of hemodialysis on RV changes? Prior studies suggest a daily but short session might produce better outcome compared to standard 3h session x 3 times per weeks? 3.2 What do you think about the effect of ultrafiltration rate in RV dimension change? 3.3 What are your suggestion to potentially address the RV/LV dimension changes and blood volume overload in those patients? 

Response 3.1-3.3: We fully agree with your suggestion that both dialysis modality and ultrafiltration rate — particularly in the context of preload sensitivity — should be considered when interpreting cardiac remodeling patterns in this population. We have merged your three insightful comments into a dedicated paragraph within the discussion section (under Parameters associated with cardiac remodeling after AVF placement). The newly added paragraph reads as follows (p. 13, line 464-476): In this context, dialysis modality and ultrafiltration rate may play a significant role in modulating cardiac remodeling. Prior randomized studies have demonstrated that daily dialysis leads to improved volume and blood pressure control, reduced interdialytic weight gain, and lower ultrafiltration volumes or rates compared to conventional thrice-weekly schedules [33,34]. These changes were associated with a significantly greater reduction in left ventricular mass (−16.4 ± 2.9 g vs. −2.6 ± 3.2 g), and reductions in both LVEDV and RVEDV (by 11% and 12%, respectively), with more pronounced effects observed in anuric patients. Even after a single hemodialysis session, preload reduction can significantly decrease LVEDV, and the right ventricle appears to be even more sensitive to such preload changes [35]. Tailored strategies—such as frequent or prolonged dialysis sessions aimed at avoiding excessive ultrafiltration rates and achieving stricter volume control—may therefore be especially beneficial in mitigating cardiac remodeling following AVF creation.
